

# Vertical profiles of wind gust statistics from a regional reanalysis using multivariate extreme value theory

Julian Steinheuer[1,2] and Petra Friederichs[3]

[1]Institute for Geophysics and Meteorology, University of Cologne
[2]Hans-Ertel Centre for Weather Research, Cologne/Bonn
[3]Institute of Geoscience, University of Bonn

**Correspondence:** Julian Steinheuer (Julian.Steinheuer@uni-koeln.de)

**Abstract.** Many applications require wind gust estimates at very different atmospheric height levels. For example, the renewable energy sector is interested in wind and gust predictions at the hub height of a wind power plant. However, numerical weather prediction models typically derive estimates for wind gusts at the standard measurement height of $10\,\mathrm{m}$ above the land surface only. Here, we present a statistical post-processing to derive a conditional distribution for hourly peak wind speed as a function of height. The conditioning variables are taken from the regional reanalysis COSMO-REA6. The post-processing is trained using peak wind speed observations at five vertical levels between $10\,\mathrm{m}$ and $250\,\mathrm{m}$ of the Hamburg Weather Mast. The statistical post-processing is based on a censored generalized extreme value (cGEV) distribution with non-stationary parameters. We use a least absolute shrinkage and selection operator to select the most informative variables. Vertical variations of the cGEV parameters are approximated using Legendre polynomials, such that predictions may be derived at any desired vertical height. Further, the Pickands dependence function is used to assess dependencies between gusts at different heights. The most important predictors are the $10\,\mathrm{m}$ gust diagnostic, the barotropic and the baroclinic mode of absolute horizontal wind speed, the mean absolute horizontal wind in $700\,\mathrm{hPa}$, the surface pressure tendency, and the lifted index. Proper scores show improvements with respect to climatology of up to $60\,\%$ especially at higher vertical levels. The post-processing model with a Legendre approximation is able to provide reliable predictions of gusts statistics at non-observed intermediate levels. The strength of dependency between gusts at different levels is non-stationary and strongly modulated by the vertical stability of the atmosphere.

## 1 Introduction

Severe wind events are one of the main weather hazards for humans and economics. Extreme wind gusts cause damage to buildings, with effects from loose flying objects to uncovering complete roofs. The hazard affects whole forests, especially those with shallow rooting trees such as spruce – the most used timber in Germany. For the energy sector, wind prediction is becoming more relevant due to the growing demand in renewable energy, and especially in wind power generation. A steady strong wind is most efficient for the power production, as the produced power at wind plants is proportional to the cube of the horizontal wind speed. The wind energy plant rotors react slowly to fluctuations in wind patterns, and thus are not able to transform the higher energy of wind gusts into electricity. On the contrary, if the shear forces due to gusts are too strong



on the rotor, they can lead to a deactivation of the complete wind park. For a stable electricity network, large wind variations are problematic and forecasts need to capture these variations. The hubs of power plants reach heights above 150 m and their size is increasing, especially in offshore parks. For the planning and operation of wind power plants, accurate estimates and forecasts of wind gusts are therefore of great value and requested not only near the surface but along their entire vertical extent.

      Regional reanalyses provide a consistent retrospective data set of the 3 dimensional state of the atmosphere. Reanaly-
ses distinguish themselves by incorporating all available observations via data assimilation in a numerical weather predic-
tion (NWP) model. The regional reanalysis COSMO-REA6 (Bollmeyer et al., 2014) represents such a high-resolution reanaly-
sis for Europe, and is currently available for the period from 1995 to 2017 [1]. The grid spacing of COSMO-REA6 is about 6 km.
It already provided guidance for renewable energy applications (e.g. Frank et al., 2019). Due to the short-time nature of
gusts – following World Meteorological Organization (2014) gusts are defined as the maximum of 3 s averaged wind
speeds – its direct simulation is not possible within a NWP model. COSMO-REA6 therefore provides a diagnostic of the
expected speed of wind gusts at a height of 10 m above the surface (Doms and Baldauf, 2011; Doms et al., 2011). Although
this estimate of the gust speed in COSMO provides valuable information for the observed gusts (Friederichs et al., 2018), it is
only given at 10 m height without an uncertainty estimate. This study is thus aiming at developing a post-processing for the
distribution of wind gusts at any height of a wind power plant.

Several approaches have been employed for the post-processing of wind and wind gusts. With the aim of applying this to
the risk assessment for off-shore wind farms, Patlakas et al. (2017) develop a deterministic post-processing based on Kalman
filtering, and Born et al. (2012) compare different gust estimates including uncertainty measures. Staid et al. (2015) pro-
pose a Gaussian forecast for maximum-value wind for offshore environments and Messner and Pinson (2019) use an adap-
tive lasso vector autoregression for forecasting wind power generation at wind farms. Probabilistic methods employ non-
homogeneous regression, e.g., Thorarinsdottir and Johnson (2012) for wind gusts, and Lerch and Thorarinsdottir (2013) for
wind speed, or ensemble model output statistics (EMOS, Scheuerer and Möller, 2015; Baran and Lerch, 2015) for wind
speed. Petroliagis and Pinson (2012) connect extreme winds with the ECMWF Extreme Forecast Index in order generate
early wind warnings. Forecasting wind gusts based on an ensemble prediction system is applied on winter storms of six years
by Pantillon et al. (2018). Friederichs et al. (2009) compare several distributions such as gamma, log-normal and generalized
extreme value distribution (GEV) for wind gusts as obtained from the observational network in Germany. They show that the
GEV is most appropriate to reliably estimate the distribution of wind gusts and theoretically consistent. Demonstrating an
evaluation method for predictive GEV distributions, Friederichs and Thorarinsdottir (2012) developed a Bayesian GEV model
for wind gusts. Post-processing for wind gusts using extreme value theory (EVT) and accounting for spatial dependencies was
developed in Friederichs et al. (2018) and Oesting et al. (2017).

In this study we propose a post-processing for the vertical structure of wind gusts at the location of the Hamburg Weather
Mast (Brümmer et al., 2012). The statistical model prediction is conditioned on the state of the atmosphere as given by the
COSMO-REA6 reanalysis (Bollmeyer et al., 2014). Our post-processing provides a predictive distribution at an arbitrary height
between 10 m and the top of the Hamburg Weather Mast, which is given in terms of parameters of a generalized extreme value

---

[1]https://www.dwd.de/DE/klimaumwelt/klimaueberwachung/reanalyse/reanalyse_node.html





distribution (GEV). Variable selection is performed with the least absolute shrinkage and selection operator (Tibshirani, 1996).

We further investigate the bivariate dependence between gusts at different heights using the Pickands dependency function.

The remainder of this article is structured as follows. In section 2 we describe the observations at the Hamburg Weather Mast and the regional reanalysis COSMO-REA6. Section 3 provides the statistical model used for the post-processing and introduces the bivariate Pickands function. The results are discussed in section 4. We end with a conclusion in section 5.

## 2   Data

### 2.1   Hamburg Weather Mast

Our targed data are hourly gusts as measured at the Hamburg Weather Mast. The Meteorological Institute at the University of Hamburg, partnered with the Max Planck Institute for Meteorology, is operating the measuring site in Hamburg, Germany (tall mast $53° 31' 9.0''$ North and $10° 6' 10.3''$ East; 10 m mast with $53° 31' 11.7''$ North and $10° 6' 18.5''$ East). The wind is measured with a 20 Hz frequency by a 3d-ultra-sonic anemometer (METEK USA-1) at $z =$ 10 m, 50 m, 110 m, 175 m, 250 m height. The

raw wind data are averaged observations over 3 s (Brümmer et al., 2012), and used to calculate hourly gusts as the maximum of raw wind data over one hour. The data cover a period of 11 years from January 1, 2004 to December 31, 2014.

### 2.2   Regional Reanalysis COSMO-REA6

The regional reanalysis COSMO-REA6 of the German Weather Service (DWD) was developed within the Hans-Ertel Centre for Weather Research (Bollmeyer et al., 2014) and provides the set of predictive variables. The reanalysis system is based

on the NWP model COSMO (Baldauf et al., 2011) and covers the CORDEX EUR-11 domain with a horizontal grid spacing of approximately 6 km (0.055°). Vertically, the reanalysis comprises of 40 layers from surface to 40 hPa. The time output resolution for the 3D-fields is one hour. The data assimilation scheme uses a continuous nudging. The Hamburg Weather Mast data are not assimilated into the COSMO-REA6. We pre-selection potentially informative covariates over a region of 25 grid box columns around the Hamburg weather mast location (more details in section 3.3).

## 3   Method

We denote the hourly gust data as $Y(z,t)$, with $z$ the height and $t$ the time. As they represent maxima of 3 s data over a block of one hour, a natural distribution to represent such block maxima is the GEV distribution. The extreme value theorem (Fisher and Tippett, 1928; Gnedenko, 1943) proves that under certain conditions the GEV is the limit distribution of the re-scaled block maxima when the block size reaches infinity.





**Theorem 1** (Extreme Value Theorem, Gnedenko (1943)). *Let $X_1, \ldots, X_n$ be a sequence of independent random variables and $M_n = max\{X_1, \ldots, X_n\}$ the block maximum. If there exist sequences of constants $\{a_n > 0\}$ and $\{b_n\}$ such that*

$$Pr\{(M_n - b_n)/a_n \leq y\} \to G(y) \quad as \quad n \to \infty, \tag{1}$$

*for a non-degenerate distribution function G, then G is a member of the GEV family with distribution function*

$$G(y; \mu, \sigma, \xi) = \begin{cases} \exp\left(-\left[1 + \xi\left(\frac{y-\mu}{\sigma}\right)\right]^{-1/\xi}\right) & \xi \neq 0 \\ \exp\left(-\exp\left[-\left(\frac{y-\mu}{\sigma}\right)\right]\right) & \xi = 0, \end{cases} \tag{2}$$

*defined on $\{y : 1 + \xi(y - \mu)/\sigma > 0\}$, where $-\infty < \mu < \infty$, $\sigma > 0$ and $-\infty < \xi < \infty$.*

In real world applications a sensible question is whether the asymptotic limit is already reached in samples of finite block size. In order to avoid biases due to non-asymptotic behavior and to concentrate on gusts above a certain level, we censor the data at a given threshold $u$ by setting $Y_u = u$ for $Y < u$ and $Y_u = Y$ for $Y \geq u$. Likewise the censored GEV (cGEV) is given as $G_u(y; \mu, \sigma, \xi) = G(y; \mu, \sigma, \xi)$ if $y \geq u$ and $G_u(y; \mu, \sigma, \xi) = 0$ otherwise. The respective density function has a density mass at $u$ that represents the probability $Pr(Y \leq u) = G_u(u; \mu, \sigma, \xi)$. This procedure is similar to the censored representation of rainfall in Scheuerer (2013) or Friederichs (2010).

## 3.1 Post-processing and verification

We thus assume that $Y(z, t)$ follows a cGEV with $G_u(y; \mu(z,t), \sigma(z,t), \xi(z,t))$, such that the parameters $\mu(z,t), \sigma(z,t), \xi(z,t)$ are non-homogeneous in space (i.e. height) and time. The non-stationarity is explained through $L$ covariates $C_l(t)$ assuming a linear regression ansatz

$$\mu(z,t) = \mu_0(z) + \sum_{l=1}^{L} \mu_l(z)C_l(t), \text{ and } \sigma(z,t) = \exp\left(\sigma_0(z) + \sum_{l=1}^{L} \sigma_l(z)C_l(t)\right). \tag{3}$$

The exponential inverse link function in Eq. (3) guarantees that the scale parameter is always positive. We further assume a Gumbel-type GEV with $\xi = 0$. The reason for this choice is discussed later in the results section 4. In order to be able to interpolate the parameters in space, we approximate their height dependence using a linear combination of Legendre polynomials up to the order $K$, namely $P_0(\eta) = 1$, $P_1(\eta) = \eta$, $P_2(\eta) = 1/2(3\eta^2 - 1)$, $\ldots$, where $\eta \in [0, 1]$ is a generalized height. Each parameter $\mu_l(z)$ and $\sigma_l(z)$ for $l = 0, \ldots, L$ is modelled as

$$\mu_l(z) = \sum_{k=0}^{K} \mu_{lk}P_k(\eta(z)), \text{ and } \sigma_l(z) = \sum_{k=0}^{K} \sigma_{lk}P_k(\eta(z)). \tag{4}$$

By including Eqs. (3) and (4) into the density formulation of $G_u(y; \mu, \sigma, \xi)$ we obtain a likelihood function for $Y$ at each level $z$ and time $t$.

The cGEV parameters are then inferred using MLE and the conditional independence assumption. In order to avoid overfitting and to assess sampling uncertainty we apply a cross-validation procedure. For each year in the time sequence the





parameter estimation is performed on a reduced data set, where the respective year of data is left out. We thus obtain one set of parameter estimates for each of the 11 years that is independent of the data of the respective year. Further, the variability of the parameter estimates provides an measure of the sampling uncertainty.

The approximation using Legendre polynomials allows for an estimation using the data at all heights simultaneously. This post-processing model is denoted as *Legendre*. In order to assess the predictability in vertical space, an additional leave-one-out procedure is applied, where the layer to be predicted is withheld during the estimation procedure. This procedure is denoted as *leave-out*. We finally also estimate the parameter for each level independently, denoted as *layer-wise*, in order to quantify how well performs the approximation of the vertical variation of the parameter using Legendre polynomials .

As the number of covariates $L$ should be restricted, we a-priori perform a selection of covariates using the least absolute shrinkage and selection operator (LASSO) as described in Tibshirani (1996). The LASSO penalizes non-zero regression parameters $\mu_{lk}$ and $\sigma_{lk}$. Depending on the a parameter $\lambda$, they are forced to zero unless they are really relevant for maximizing the likelihood. For a given log-likelihood function $l(\Theta)$, where the vector $\Theta$ contains all unknown parameters, the LASSO approach maximizes

$$l_\lambda(\Theta) = l(\Theta) - \lambda \sum_{l=1}^{L} \sum_{k=0}^{K} (|\mu_{lk}| + |\sigma_{lk}|). \tag{5}$$

The larger $\lambda$ the stronger is the penalization and the more regression parameters become zero. The constant parameters $\mu_{0k}$ and $\sigma_{0k}$ are not penalized, and thus a large shrinkage parameter $\lambda$ results in a stationary cGEV model.

The verification of the cross-validated predictive distribution is performed using proper scoring rules (Gneiting and Raftery, 2007). We use the quantile score (QS) for predictive quantiles $q_\tau = cGEV^{-1}(\tau; \mu, \sigma, \xi)$ of the censored data at the probability $\tau$

given as

$$QS_\tau(q_\tau, y_u) = \tau(q_\tau - y_c) I_{y_u \le q_\tau} + (\tau - 1)(q_\tau - y_u) I_{y_u > q_\tau}, \tag{6}$$

following (Friederichs and Hense, 2007) and its decomposition (Bentzien and Friederichs, 2014). The observation $y_u$ is also censored with $y_u = y$ for $y \ge u$ and $y_u = u$ otherwise. We further use the Brier score (BS, Brier, 1950) and the continuous ranked probability score (CRPS, Hersbach, 2000) for the cGEV. The CRPS is proportional to the integral of the QS over all

probabilities $\tau$ (Gneiting and Raftery, 2007) or the BS over all thresholds (Hersbach, 2000). Skill measures are provided as the percentage improvement of the scores with respect to a reference forecast. A typical reference is the unconditional (i.e. stationary) distribution – here in terms of a censored GEV for each level individually – of the observed gusts at each mast level. All scores are evaluation using censoring. Proper scoring rules can be decomposed into contributions related to reliability and resolution. We use the decomposition for the QS as developed in Bentzien and Friederichs (2014).

For the calculations, we used the Statistical Programming Language R (R Core Team, 2016) with modified routines from the packages *ismev* (for estimation, Heffernan and Stephenson, 2016) and *verification* (for validation, NCAR - Research Applications Laboratory, 2015).



## 3.2 Residuals and spatial dependence

Residuals of the gust observations are derived using the cross-validated cGEV parameter estimates to transform the data to
a standard GEV (e.g standard Gumbel with $\mu = 0, \sigma = 1, \xi = 0$). No censoring is applied to calculate the residuals, i.e. we assume, that the GEV using the fitted cGEV parameters represent the gust values also below the threshold $u$. A quantile-quantile plot (QQ plot) is used to assess the validity of this assumption.

Another assumption that is explicitly used in the MLE is the conditional independence of the gust observations at the different mast levels. Although this assumption mainly concerns the uncertainty of the parameter estimates, conditional dependence will
become relevant if one would like to draw realizations of the vertical gusts or derive aggregated measures (e.g. the probability of observing a gust at any level of the mast). To assess dependence of the gusts between different height levels, we use the bivariate Pickands dependence function (Pickands, 1981). The bivariate extreme value distribution for standard Fréchet variables ($\mu = \sigma = \xi = 1$) has the form

$$G(y_1, y_2) = \exp\left(-\left(\frac{1}{y_1} + \frac{1}{y_2}\right) A(\omega)\right), \tag{7}$$

with $\omega = y_2/(y_1 + y_2)$, and hence $\omega \in [0,1]$. The Pickands dependence function $A(\omega)$ describes the dependency of a pair of random variables $(Y_1, Y_2)$ with standard Fréchet margins. A non-parametric estimate of $A(\omega)$ is given in Pickands (1981) with

$$A_m^P(\omega) = m \left[\sum_{i=1}^{m} \min\left(\frac{1}{y_{1,i}\omega}, \frac{1}{y_{2,i}(1-\omega)}\right)\right]^{-1}, \tag{8}$$

for $m$ pairs of observations. Here we use a modification to approach convexity by Hall and Tajvidi (2000)

$$A_m^{HT}(\omega) = m \left[\sum_{i=1}^{m} \min\left(\frac{\bar{y}_1}{y_{1,i}\omega}, \frac{\bar{y}_2}{y_{2,i}(1-\omega)}\right)\right]^{-1}, \tag{9}$$

with $\bar{y}_j = m(\sum_{i=1}^{m} 1/y_{i,j})^{-1}$. $A_m^{HT}(\omega)$ is used as a limiting function. A convex and therefore valid Pickands dependence function is given by the convex minorant $A_m^{HT,c}$ of $A_m^{HT}(\omega)$ (i.e. the largest convex function on $[0,1]$ that has no values exceeding $A_m^{HT}(\omega)$). The R package *evd* (Stephenson, 2018) provides the routines to estimate the function.

## 3.3 Preparation of covariates

We consider the following variables as covariates: the wind gust diagnostic at $10\,\mathrm{m}$ (VMAX_10M), the vertical profile of the horizontal wind speed at mast levels, the horizontal (Vh_700) and veritical (W_700) wind speed at 700hPa, surface pressure tendency ($d_t$ P), lifted index (LI), total water content (TWATER), atmospheric temperature in 2m height (T_2M), tendency in convectively available potential energy ($d_t$ CAPE), vertical shear of horizontal wind between $6\,\mathrm{km}$ and $1\,\mathrm{km}$ (Vh_SHEAR), the temporal variance of VMAX_10M (VAR$_t$ VMAX_10M) and the phase of the annual cycle. For a summary of the description
below see Table 1. All covariates are standardized before they enter the cGEV regression model.

The gust diagnostic in COMSO-REA6 is probably the most informative variable, since it aims as an estimate of the potential strength of a gust near the surface. Gusts are generated, on the one hand, by turbulent deflection of upper air wind to





the surface (Brasseur, 2001) and, on the other hand, by convective downdrafts (Nakamura et al., 1996). The turbulent gust diagnostic in COSMO-REA6 is given by an empirical relation to the $10\,\mathrm{m}$ wind velocity and the surface drag coefficient for
momentum (Schulz and Heise, 2003; Schulz, 2008). The convective gust diagnostic depends on the downdraft formulation in the convection scheme (Schulz and Heise, 2003) and includes the height and the kinetic energy of the downdraft. VMAX_10M is the maximum of the turbulent and convective gust diagnostic. The differences between the observed gusts at $10\,\mathrm{m}$ height at the Hamburg Weather Mast and the COSMO-REA6 gust diagnostics are displayed in Fig. 1. The differences have a negative bias of about $-1.03\,\mathrm{ms}^{-1}$, i.e. COSMO-REA6 slightly overestimates the strength of the gusts. The standard deviation amounts
to about $1.8\,\mathrm{ms}^{-1}$. We also include the variance of VMAX_10M over 5 hours (Var$_t$ VMAX_10M) as a covariate.

Since gusts are naturally related to mean wind speed, we include the horizontal velocities at the station location. COMSO-REA6 has a staggered grid, so the wind velocity is given as the absolute velocity of the centered zonal and meridional velocities. To represent the state of the local vertical profile of the horizontal wind velocity in a height independent variable, we use a principal component analysis. A principal component analysis of the wind velocity at the different heights reveals, that most
variability (about 92 %) is explained by a mode of variability where all wind anomalies have the same sign, with a slight increase in variability at higher levels. The second mode of variability, which explains about 6% of the total variability, represents a dipole (i.e. baroclinic) structure with positive anomalies at the upper two levels and corresponding negative anomalies at the lower most 3 levels. The latter mode is called the baroclinic wind mode (Vh_EOF2), while the former – although not completely barotropic – is called the barotropic wind mode (Vh_EOF1).

An important index to capture vertical instability is the lifted index (LI, e.g. Bott, 2016). It is defined as the difference between the temperature in $500\,\mathrm{hPa}$ and the temperature of an air parcel that is adiabatically lifted up from the surface to $500$ hPa. Negative values indicate a potentially unstable atmosphere, which could lead to convection and hence gusts. If convection takes place, CAPE is consumed and a tendency in CAPE is seen in the reanalysis data. Thus we include the tendency of CAPE (d$_t$ CAPE) over one hour as a covariate. We further use the total water content (TWATER) of the column
that includes the location of the Hamburg weather mast. All these covariates are calculated for the vertical column of the grid point closest to the mast location.

We further include information on the atmospheric circulation above the boundary layer at 700 hPa surrounding the Hamburg weather mast. The wind velocities at the closest 25 grid cells are used to calculate an averaged horizontal (Mean$_h$ Vh_700) and vertical (Mean$_h$ W_700) wind speed as well as the respective standard deviations over that region SD$_h$ Vh_700, and SD$_h$ W_700,
respectively. Another possible indicator for gust activity is the tendency of pressure at the surface over one hour within the area surrounding the weather mast. The pressure tendency d$_t$ P is an averaged tendency again over the 25 nearest grid points.

The annual cycle is represented by a linear combination of a sine and cosine function with a period of one year (AC_COS and AC_SIN).



## 4 Results

Several decisions are needed before setting up the post-processing. The first concerns the threshold for censoring. We choose the climatological 50 %-quantile estimated at each level, respectively, which corresponds to $5.79\,\mathrm{ms}^{-1}$ (at $10\,\mathrm{m}$ height), $7.40\,\mathrm{ms}^{-1}$ ($50\,\mathrm{m}$), $8.65\,\mathrm{ms}^{-1}$ ($110\,\mathrm{m}$), $9.69\,\mathrm{ms}^{-1}$ ($175\,\mathrm{m}$), and $10.54\,\mathrm{ms}^{-1}$ ($250\,\mathrm{m}$). We further decide to fix the shape parameter $\xi$ to zero for two reasons. Studies of wind gusts often reveal a negative $\xi$ for the fitted GEV (e.g. Friederichs et al., 2009), i.e. a Weibull-type GEV with an upper endpoint. Any future gust above this endpoint would have predictive probability zero,

which would results in a very bad forecast. A Gumbel-type GEV therefore reduced the risk of missing an extreme gust. The second argument is the stability of the maximum likelihood optimization. The estimation of $\xi$ introduces large uncertainties. Particularly with a large amount of parameters (i.e. covariates) the optimization procedures is often stuck in a local maximum. This is particularly critical, if the domain of the distribution is restricted, as is the case for a Weibull-type GEV. Finally, to approximate the vertical variation of the cGEV parameters we use the first three Legendre polynomials $P_0$ (constant), $P_1$ (linear),

and $P_2$ (quadratic). Higher order polynomials did not provide any added value (not shown).

### 4.1 Model selection

The next step is the selection of most important predictors. The variable selection is performed using the LASSO with a relatively small penalization ($\lambda = 0.02 \times m$, where $m$ is the number of included observations). LASSO estimates are derived for each cross-validated sample, respectively, providing eleven sets of penalized regression coefficients. Since the covariates

are standardized, the absolute value of each related coefficient is proportional to the importance of the covariate. We select a covariate, if at least one of its three Legendre coefficients is consistently below or above zero for all 11 cross-validation samples. If a covariate is selected, we allow for the full flexibility in the vertical including all three Legendre polynomials, since especially the higher order polynomials are very sensitive to the penalization.

Table 2 represents the regression coefficients obtained for the Legendre model with the selected covariates but without pe-

nalization. The parameters that resisted the penalization are displayed with bold numbers. If no regression coefficient is given in Table 2, then the covariate was not selected. For the location parameter $\mu$ the most informative covariate is in general the barotropic wind mode (Vh_EOF1) and the gust diagnosis (VMAX_10M). The averaged horizontal wind (Mean$_h$ Vh_700) provides some additional information. Similarly important are the pressure tendency (d$_t$ P) with a positive pressure tendency (e.g. a passing cold front) being related to in increase in gust activity and TWATER with a negative regression coefficient.

The influence of the covariates on $\sigma$ is generally weaker than on $\mu$. Here the most informative covariate is indeed VMAX_10M leading to an increase in $\sigma$ if VMAX_10M is large. The variance of the predictive cGEV is significantly increased if Var$_t$ VMAX_10M is large. We discuss the influence of Var$_t$ VMAX_10M later in this section. Vh_EOF1 was not selected by the LASSO approach, but some additional information is provided by the baroclinic wind mode (Vh_EOF2). The weak influence of AC_COS indicates a slight increase in gust activity during summer, that is not explained by the other covariates.

The interpretation of the role of the covariates is not straight forward, since the selected covariates are correlated. This is particularly the case for the $10\,\mathrm{m}$ gust diagnostic and the barotropic wind mode. The omission of one would therefore lead to





a modified role of the other. The most important covariates, notably the wind covariates, roughly reveal that stronger winds results in increased $\mu$ and $\sigma$ parameters of the cGEV. Further, there is a remarkable influence by integrated water content and the pressure tendency. A positive pressure tendency is associated with stronger wind gusts, and one may argue that the probability of gusts is increase during the passage of a cold front. The role of TWATER is at first less obvious. TWATER shows a pronounced annual cycle, since the warmer atmosphere during summer has a larger water vapor capacity. Likewise, gusts are on average stronger during winter than during summer. The mean $10\,\mathrm{m}$ wind gust at the Hamburg Weather Mast is about $6.3\,\mathrm{ms}^{-1}$ in winter and $5.78\,\mathrm{ms}^{-1}$ in summer. One should thus be careful in the interpretation, as the negative relation between TWATER and gustiness may just be a consequence of the annual cycle, and should not be interpreted as a causal relation.

The covariate $\mathrm{Var}_t$ VMAX_10M was not included in an earlier version of the Legendre model. Figure 2 (a) shows the residuals using the Legendre model without $\mathrm{Var}_t$ VMAX_10M against the observed gusts. The highest gusts above $20\,\mathrm{ms}^{-1}$ are well captured since the residuals are generally small with values between $-1$ and $4$. However, the QQ-plot in Fig. 2 (b) indicates the three outliers that are not well captured by the model. The outliers correspond to gusts of about 15 to $20\,\mathrm{ms}^{-1}$ and are therefore of relevance. Two of them occur on 26 August 2011. Figure 3 (a) shows the model predictions on 26 August 2011. The predictive quantiles are calculated using a GEV with the Legendre estimates of the cGEV. The outliers are observed at 18 CET and 20 CET and are well exceeding the predictive $99\,\%$-quantiles, whereas COSMO-REA6 diagnoses a gust of about $20\,\mathrm{ms}^{-1}$ at 19 CET. The observed gusts are related to two convective storms that passed over Hamburg. The COSMO-REA6 analysed a convective cell over Hamburg but with an incorrect timing. The adjusted prediction including $\mathrm{VAR}_t$ VMAX_10M is shown in Fig. 3 (b). We now see an increase in the predicted range of the gusts such that the observed gusts are within the 99% range of the prediction. The QQ-plot of the Legendre model including $\mathrm{VAR}_t$ VMAX_10M (Fig. 4 (a)) shows that the two outliers on 26 August 2011 are now eliminated at the cost that the Legendre model now slightly overestimates the high quantiles. With the inclusion of the temporal variability of the $10\,\mathrm{m}$ gust diagnostic, we improved the the post-processing model mainly through increasing the $\sigma$ parameter when gustiness in the reanalysis strongly varies over time. The role of this covariate is thus to account for timing errors in the reanalysis, which might be particularly large for weather situations that favour small convective cells. This method successfully eliminates two of the three outliers. Figure 4 (b) shows the QQ-plot at $110\,\mathrm{m}$. The remaining outlier is also present at a higher level, but the overestimation of the high quantiles is much weaker than in $10\,\mathrm{m}$.

## 4.2 Verification

The post-processing is assessed using proper verification skill scores. We first assess the effect of the Legendre approximation. Figure 5 (a) – (c) shows skill scores of the layer-wise model with climatology as a reference. The $99\,\%$-QSS indicates remarkable improvements of about $45\,\%$ to $60\,\%$ with respect to climatology. The BSS evaluates the predictive probability increasing the climatological $99\,\%$-quantile in each level, respectively. The BSS is smaller than the QSS with values ranging from about $10\,\%$ in the lowest level to $40\,\%$ in $250\,\mathrm{m}$. The CRPSS ranges between $40\,\%$ and $50\,\%$. Ideally, an approximation of the vertical variation of the cGEV parameters by Legendre polynomials should not deteriorate the skill scores. Figure 5(d) – (f) show the skill score of the Legendre model with the layer-wise model as reference. The reduction in skill is





not larger than 7 %, being largest in the QSS and BSS at the 10 m level. We conclude that the Legendre model represents an appropriate for all layers.

The advantage of the Legendre model is the possibility to provide predictions at levels where no observations are available. Figure 6 (a) – (c) represents the skill score for the leave-out model with climatology as reference. All skill scores show a strong decrease in skill at 10 m and 250 m. At 10 m, the BSS even shows negative skill. In Fig. 6 (d) – (f) the direct comparison visualize that except for the lowest and highest level the loss in skill is only of about 10 % at the most when compared to what is obtained with the layer-wise model. The decomposition of the QSS of the 99 %-quantiles at 10 m and 110 m shows that the loss in predictive skill is mainly due to the reliability term, while resolution remains almost constant. It further shows, that the reliability is particularly bad for the leave-out model at 10 m height. Thus, the interpolation of the cGEV parameters is applicable, while an extrapolation to the 10 m and the 250 m level fails to provide a reliable predictive distribution.

The post-processing aims at an improved 10 m wind gust diagnostic. In order to compare the post-processed gust distribution with the COSMO-REA6 gust diagnostic, we calculate the median of a GEV using the cGEV parameters of the layer-wise model. Thereby, we calculate the median for the layer-wise model. Figure 8 shows the histogram of differences between the observations and the mean at 10 m. Compared to the gust diagnostic of COSMO-REA6 in Fig. 1, we see an improvement as the bias nearly vanishes and the standard deviation of the differences is reduced to 1.57 ms$^{-1}$. Large differences still occur in situations, where the reanalysis is not able to simulate small scale convective cells correctly in terms of timing or location.

### 4.3 Application and bivariate dependency

To illustrate the post-processing using the Legendre model, we have a closer look at the storm Emma between 29 February and 1 March 2008. During Emma we observe the largest gusts at 10 m within the complete observation period of the Hamburg Weather Mast, with 28.07 ms$^{-1}$ on 1 March 2008 between 12 and 13 CET. The storm hit a large region in Europe. In Hamburg, a storm surge flushed parts of the city. COSMO-REA6 has difficulties to exactly capture the evolution of the storm over Hamburg (Fig. 9). As the reanalysis, the post-processing misses the highest gusts on Saturday, 1st May 2008, although the prediction is provided with reasonably high uncertainties. A better prediction is generated by the post-processing on 28 Friday February 2008. By way of example, we have selected 3 hours that represent differently stratified atmosphere as indicated by vertical lines in Fig. 9. According to Bott (2016), we characterize the atmosphere as stable if LI $\geq 6$ K, as neutral if $6$ K $\geq$ LI $\geq -2$ K, and as unstable if $-2$ K $\geq$ LI. Figure 10 shows the corresponding vertical profiles of the predictive GEV distribution. In all cases the median prediction is in good agreement with the observations. On 29 February 2008, 10 CET (stable atmosphere), the observed gusts are within the inner quartile range of the predictive GEV, and slightly below the censoring threshold. The variance of the predictive GEV is small. On 1 March 2008, 1 CET (neutral atmosphere) the inner quartile range is larger, the vertical variation of the gusts is also larger and well captured by the predictive GEV. On 1 March 2008, 4 CET, the atmosphere is highly unstable. The observed gust are very close to the median of the predictive GEV. Note that LI only influences the cGEV scale parameter, and that the regression coefficient is small (see table 2).

Figure 10 suggests that the gusts do not vary independently of each other. In order to investigate the spatial dependency, we calculate the bivariate Pickands dependency function following Eq. 9. Transformation to standard Fréchet is performed





using the parameters for a climatological cGEV (i.e. assuming a stationary marginal cGEV independently in each height) and from the Legendre model (i.e. accounting for non-stationarity by post-processing). Figure 11 shows the estimated Pickands dependence function between the gust residuals in $10\,\mathrm{m}$ and $110\,\mathrm{m}$ separately for the stable, neutral and unstable cases. Using stationary marginals, the dependence between the gusts in the two levels is strong and seems independent of the stability of the atmosphere. Post-processing strongly reduces vertical dependencies in the residuals. Weakest dependence is observed in a

stable atmosphere, while in an unstable atmosphere, dependence for the post-processed residuals is almost as strong as for the climatological residuals. Variation in the dependency structure is reasonable, as the more unstable the atmosphere, the more vertical mixing is induced.

    The dependence between residual gusts at $10\,\mathrm{m}$ and higher levels decreased with distance in the vertical as indicated by the value of the Pickands dependency function at $\omega = 1/2$ in Fig. 12 (a). Again, for the climatological residuals, dependence is

strong and decreases weaker with distance than for the post-processed residuals. The decrease in dependence with distance is largest during cases with a stable atmosphere. A simple relation between the strength of dependency and the distance between layers is not given, as e.g. the dependence between gusts at $110\,\mathrm{m}$ and $250\,\mathrm{m}$ is stronger than between gusts at $110\,\mathrm{m}$ and $10\,\mathrm{m}$ (Fig. 12 (b)).

## 5   Conclusions

This study presents a post-processing of for hourly wind gusts at different vertical heights as given by observations at the Hamburg Weather Mast. The post-processing model is based on a conditional censored Gumbel-type GEV distribution. The censoring threshold is defined as the climatorlogical 50% quantile at each mast level, respectively. The censoring approach performs well and leads to a good representation of the larger gusts.

    A LASSO approach is used to select the most informative covariates. The selected variables are the COSMO-REA6 wind

gust diagnostic at $10\,\mathrm{m}$ and its temporal variance, the barotropic and baroclinic mode of absolute horizontal wind speed, the mean absolute horizontal wind in 700 hPa, the pressure tendency, the lifted index, and the grid column water content. The predictive cGEV median provides an improved gust estimate when compared to the reanalysis gust diagnostic at $10\,\mathrm{m}$.

    Vertical variations of the cGEV parameters are approximated using the three lowest order Legendre polynomials. Although best scores are obtained if the post-processing is performed for each level independently, the unified description only results

in a slight degradation of skill at the intermediate layers. The unified description induces a small bias at $10\,\mathrm{m}$ with gusts being slightly overestimated. Extrapolation of the cGEV parameters towards the $10\,\mathrm{m}$ and the upper most level generates large biases and thereby degrades skill. In contrast, interpolation towards intermediate levels is very successful, since the degradation in terms of predictive skill is barely significant when excluding the model level. The post-processing therefor not only provides calibrated predictive distributions of gusts at the observed levels, but also at arbitrary heights of the weather mast.

The strength of spatial dependency of gusts is assessed using the Pickands dependence function. The gusts at the different heights are highly dependent. Conditioning the gusts on the COSMO-REA6 covariates reduces the dependency of the residuals between heights. This reduction in dependence is significantly modulated by the stability of the atmosphere as given by the





lifted index in the sense that an unstable atmosphere increases mixing and thereby dependency. Dependency is not simply a function of distance. For a full spatial model description of the gusts, dependency needs to be modelled as a function of

atmospheric condition as well as height.

   The post-processing model as estimated for the Hamburg Weather Mast should in principle be transferable to other locations. Transferability should be assessed at the few other mast locations in Germany. However, at many locations only measurements of the 10 m are available, and it would be of interest to assess how well estimates of gust statistics are based on observations at 10 m, only. This would enable to provide estimates of vertical gust statistics at any location of the domain the COSMO-REA6

reanalysis.

*Acknowledgements.* We are gratefully to the Meteorological Institute at University of Hamburg in person of Ingo Lange for the provision of the wind data of the Hamburg Weather Mast. Special thanks go to Sebastian Buschow for helpful discussions and valuable ideas.





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




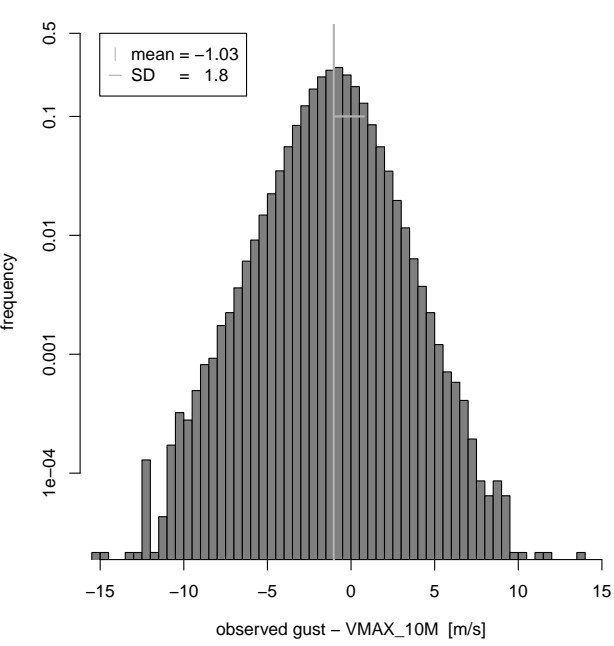

**Figure 1.** Histogram of differences between observed gusts at 10 m and the COSMO-REA6 10 m gusts diagnostic.

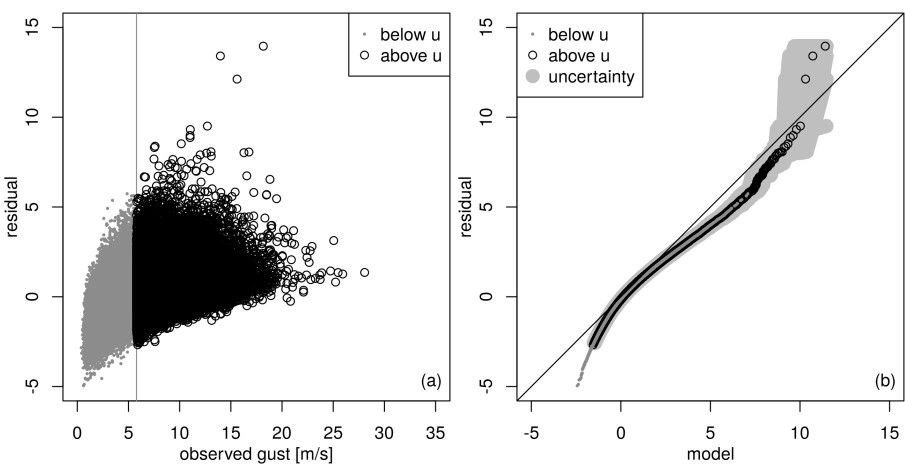

**Figure 2.** Diagnostics for Legendre model without VAR$_t$ VMAX_10M and threshold $u = 5.79\,\mathrm{ms}^{-1}$ at 10 m: (a) Scatter plot of the standard Gumbel residual against observed gusts, and (b) QQ plot of the residuals against the standard model. Uncertainty is given in light gray as the range of a 100 member bootstrap sample generated with blocks of 10 consecutive days.

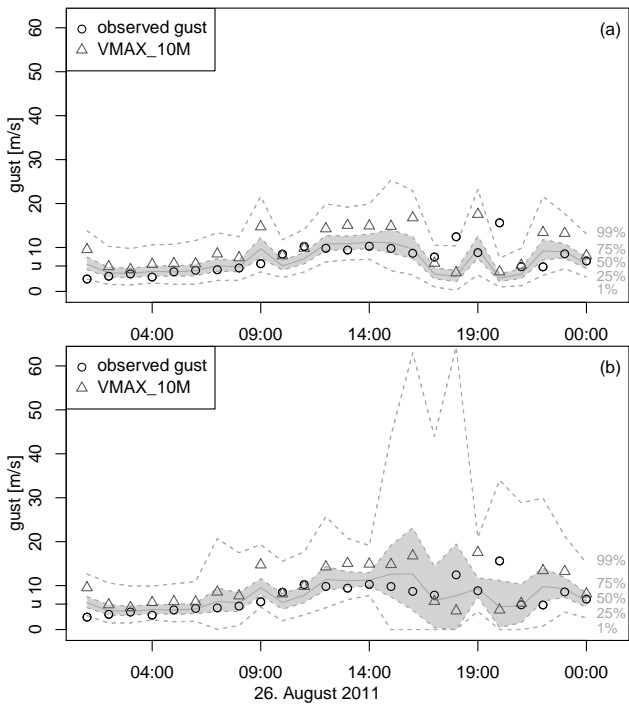

**Figure 3.** Post-processing of gusts on 26. Augsut 2011 at 10 m, (a) Legendre model without $\mathrm{VAR}_t$ VMAX_10M; (b) Legendre model. Shading indicates the predictive inner quartile range, gray line the median and dashed lines the 1% and 99% quantile. The observed gust are shown as circles, the 10 m gust diagnostic as triangles.



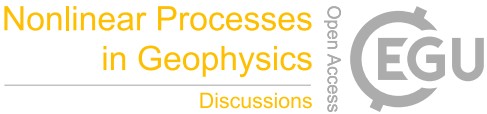
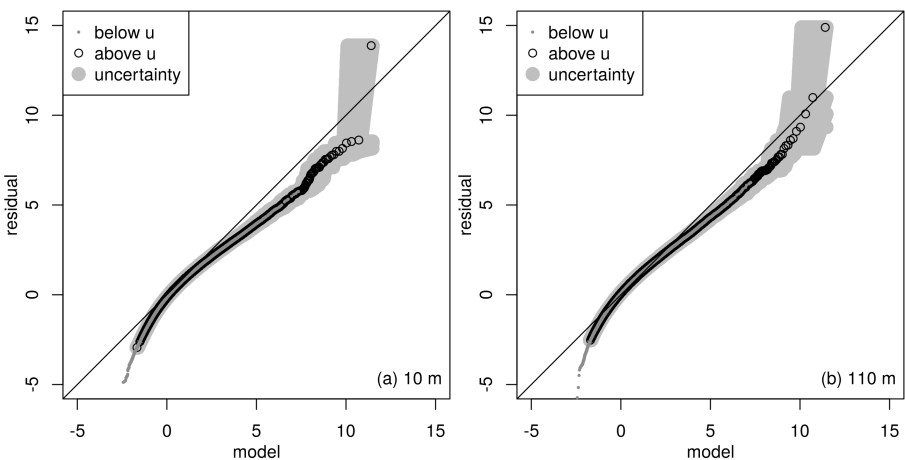

**Figure 4.** QQ plots for Legendre model (a) at 10 m and (b) at 110 m with bootstrap uncertainty as in Fig. 2.



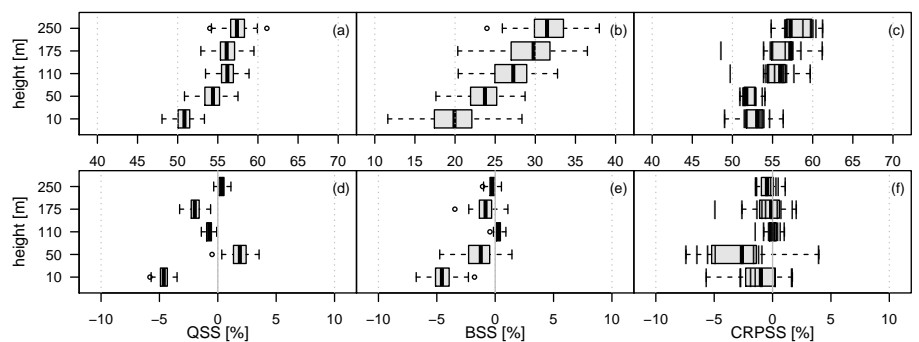

**Figure 5.** Verification skill scores for Legendre model against climatology (a) to (c) and against layer-wise model (d) to (f). The QSS is given for the predictive $\tau = 99\%$ quantile in (a) and (d); the BSS for thresholds corresponding to the climatological 99% quantile ($u =14.8$ m/s in 10 m, $u =19.26$ m/s in 50 m, $u=21.01$ m/s in 110 m, $u =22.55$ m/s in 175 m, $u =23.97$ m/s in 250 m) in (b) and (e); and the CRPSS in (c) and (f). For QSS and BSS the box-whiskers represent the 100 member bootstrap sample, with the box giving the inner quartile range. The range of the whiskers is maximal 1.5-times the width of the box. For the CRPSS the boxes represent the eleven cross-validated estimates.



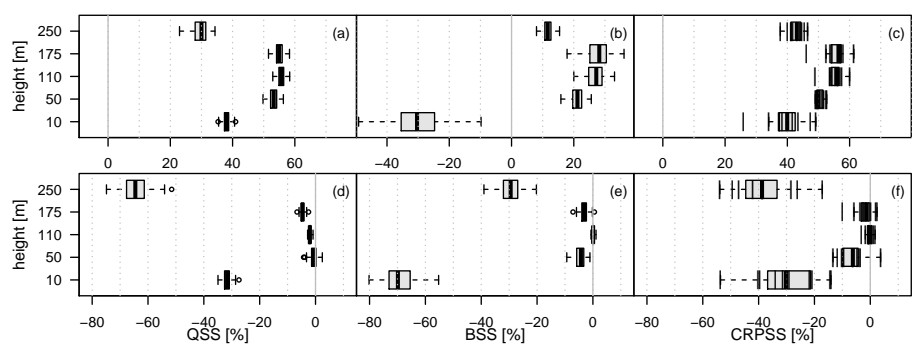

**Figure 6.** Same as in Fig. 5 but for (a) to (c) the leave-out model against climatology and (d) to (f) against layer-wise model.


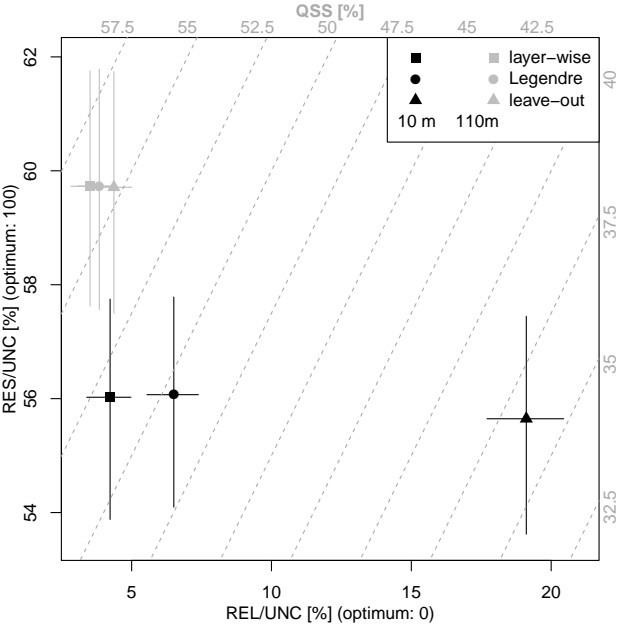

**Figure 7.** Decomposition of the QSS of the predictive 99% quantile at 10 m (black) and 110 m (gray) into scaled resolution (RES/UNC) and scaled reliability (REL/UNC) for the layer-wise, Legendre and leave-out models. The crosses show the range of the 100 member bootstrap samples. The gray dashed lines indicate the QSS. The amount of the QSS is given at the upper and right axes as gray numbers.


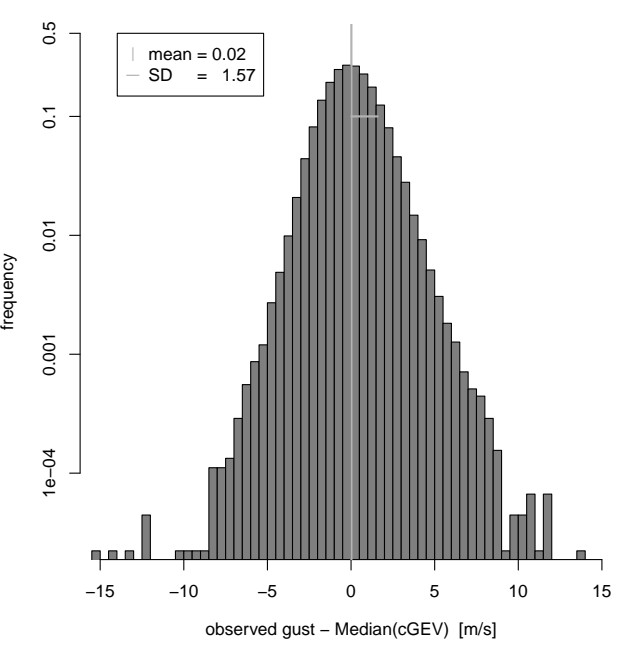

**Figure 8.** Histogram of differences between observed gusts at 10 m and the GEV median prediction at 10 m.

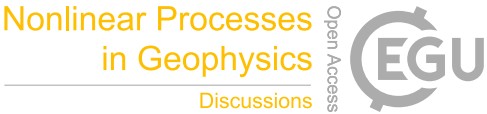

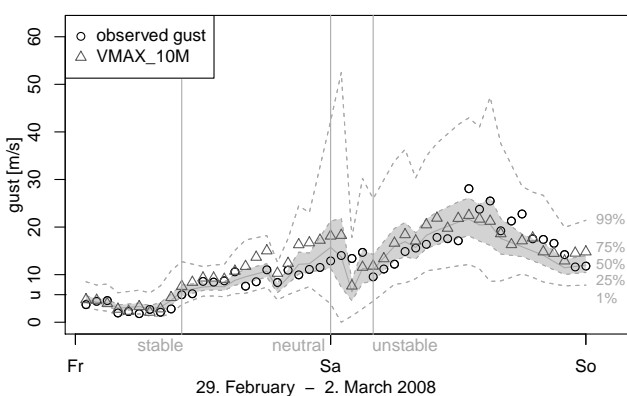

**Figure 9.** Post-processing of gusts on 29. February 2008 and 1. March 2008 at 10 m. Shading indicates the predictive inner quartile range, gray line the median and dashed lines the 1% and 99% quantile. The observed gust are shown as circles, the 10 m gust diagnostic by triangles. The vertical lines indicate times with stable (LI = 8.7 K), neutral (LI = 2.4 K), and unstable (LI = -3.1 K) conditions.



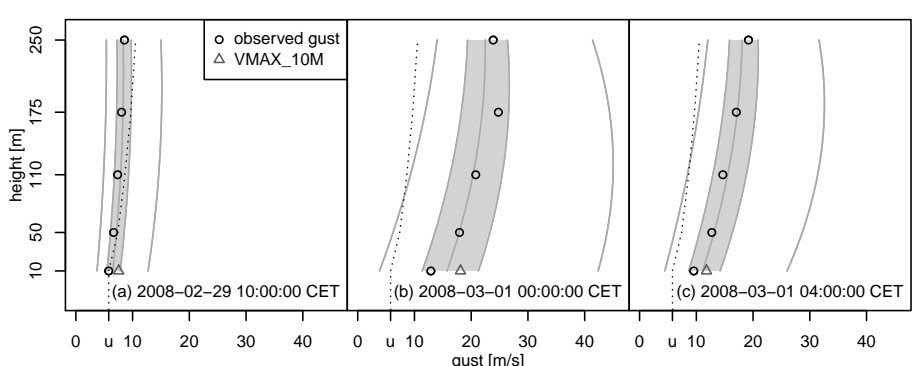

**Figure 10.** Vertical post-processing of gusts using Legendre model for times as highlighted in Fig. 9. The gray solid lines indicates the conditional quantiles using a GEV at probabilities 1 %, 25 %, 50 %, 75 %, and 99 %. The dotted line represents the censoring threshold.





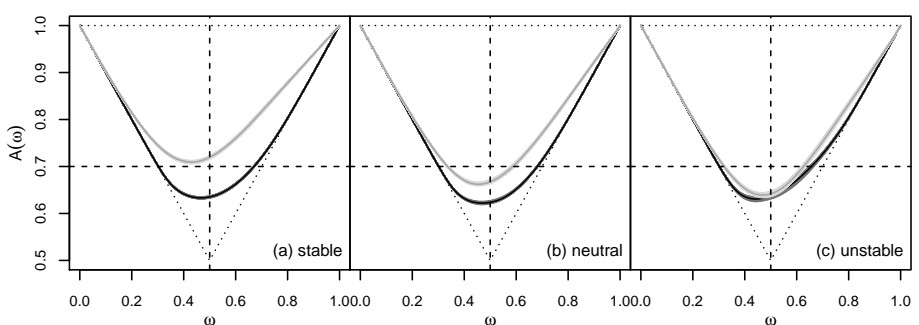

**Figure 11.** Pickands dependence function of $10\,\mathrm{m}$ and $110\,\mathrm{m}$ for Legendre model (lightgrey) and climatology (darkgrey). According the LI, the data are classified in 53 % stable (a), 36 % neutral (b), and 11 % unstable cases. Uncertainly is derived using block bootstrapping. A horizontal line at 0.7 is displayed for visualization purpose only. The dotted lines indicate complete independence with $A(\omega) = 1$ as well as complete dependence.





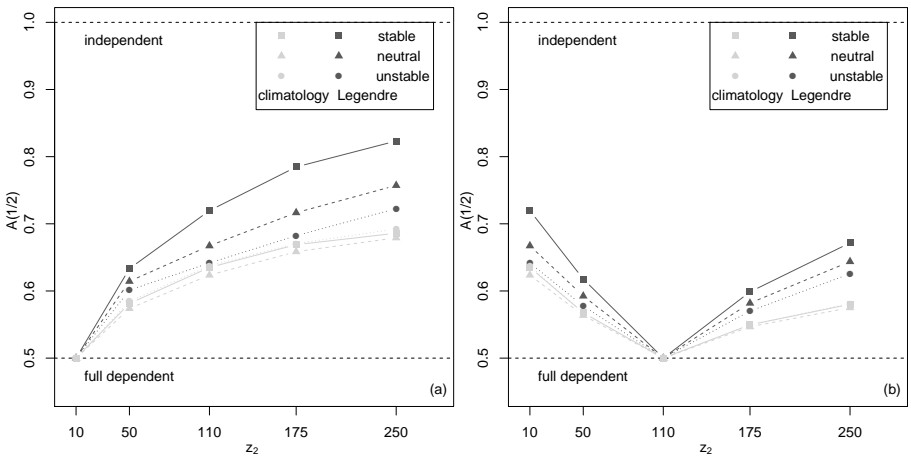

**Figure 12.** Pickands dependence function at $\omega = 1/2$ between gusts (gray) and residuals (black) at all layers and (a) $z_1 = 10\,\mathrm{m}$, and (b) $z_1 = 110\,\mathrm{m}$ for stable, neutral and unstable case as in Fig. 11.





**Table 1.** List of pre-selected covariates from the COSMO-REA6 reanalysis.

| Acronyms | Variable | Description |
|---|---|---|
| VMAX_10M | Wind gust diagnostic at 10 m | Grid value |
| $VAR_t$ VMAX_10M | Temporal variance of VMAX_10M | Variance of five consecutive ($\pm 2\,h$) grid values |
| Vh_EOF1 | Barotropic mode of absolute horizontal wind at lowest layers | Principal component of first eigenvector of covariance matrix from wind time series (11 years) at lowest 300 m (six layers) |
| Vh_EOF2 | Baroclinic mode of absolute horizontal wind at lowest layers | Principal component of second eigenvector of covariance matrix from wind time series (11 years) at lowest 300 m (six layers) |
| $Mean_h$ Vh_700 | Mean absolute horizontal wind at 700 hPa | Mean of 25 mast-surrounding grid values at layer 23 |
| $SD_h$ Vh_700 | Standard deviation of absolute horizontal wind at 700 hPa | Standard deviation of 25 mast-surrounding grid values at layer 23 |
| $Mean_h$ W_700 | Mean vertical wind at 700 hPa | Mean of 25 mast-surrounding grid values at layer 23 |
| $SD_h$ W_700 | Standard deviation of vertical wind at 700 hPa | Standard deviation of 25 mast-surrounding grid valuesat layer 23 |
| $d_t$ P | Surface pressure tendency | Mean difference between current and previous surface pressure from mast-surrounding grid values |
| LI | Lifted Index | Difference between the temperature at 500 hPa (layer 18) and the temperature of an adiabatically lifted surface air parcel |
| TWATER | Water content | Water content of the mast-including grid column |
| $d_t$ CAPE | CAPE tendency | Difference between current and previous CAPE of the mast-including grid column |
| Vh_SHEAR | Horizontal wind shear | Difference between absolute horizontal wind in 6 km (layer 17) and 1 km (layer 30) |
| T_2M | Temperature at 2 m | Grid value |
| AC_COS | Annual cosine cycle | Cosine oscillation with 1 year period |
| AC_SIN | Annual cosine cycle | Sine oscillation with 1 year period |


**Table 2.** Estimates of the regression coefficients using the Legendre model with $K = 2$. Estimates are derived without penalization including the selected covariates. Mean and standard deviation are derived from the 11 estimates using cross-validation. Bold estimates indicate the parameters, that resisted the LASSO penalization. No values are given, if the variable in not included in the Legendre model.

| Covariates | $P_0(\eta) \sim$ constant | | $P_1(\eta) \sim$ linear | | $P_2(\eta) \sim$ quadratic | |
|---|---|---|---|---|---|---|
| | $\mu_{l0}$ | $\sigma_{l0}$ | $\mu_{l1}$ | $\sigma_{l1}$ | $\mu_{l2}$ | $\sigma_{l2}$ |
| **VMAX_10M** | $\mathbf{1.23 \pm 0.01}$ | $\mathbf{0.22 \pm 0.00}$ | $-0.45 \pm 0.01$ | $-0.02 \pm 0.00$ | $0.02 \pm 0.01$ | $0.00 \pm 0.00$ |
| **Var$_t$ VMAX_10M** | | $\mathbf{0.11 \pm 0.00}$ | | $\mathbf{-0.03 \pm 0.00}$ | | $0.00 \pm 0.00$ |
| **Vh_EOF1** | $\mathbf{2.16 \pm 0.01}$ | | $\mathbf{1.11 \pm 0.01}$ | | $-0.29 \pm 0.00$ | |
| **Vh_EOF2** | $0.00 \pm 0.01$ | $\mathbf{0.10 \pm 0.00}$ | $\mathbf{0.40 \pm 0.01}$ | $0.03 \pm 0.00$ | $0.04 \pm 0.00$ | $0.00 \pm 0.00$ |
| **Mean$_h$ Vh_700** | $\mathbf{0.44 \pm 0.02}$ | $\mathbf{0.07 \pm 0.00}$ | $\mathbf{0.26 \pm 0.01}$ | $-0.01 \pm 0.00$ | $-0.00 \pm 0.00$ | $-0.01 \pm 0.00$ |
| SD$_h$ Vh_700 | | | | | | |
| Mean$_h$ W_700 | | | | | | |
| **SD$_h$ W_700** | | $\mathbf{0.04 \pm 0.00}$ | | $0.02 \pm 0.00$ | | $-0.01 \pm 0.00$ |
| **d$_t$ P** | $\mathbf{0.41 \pm 0.01}$ | $\mathbf{0.04 \pm 0.00}$ | $0.09 \pm 0.00$ | $-0.02 \pm 0.00$ | $-0.06 \pm 0.00$ | $0.00 \pm 0.00$ |
| **LI** | | $\mathbf{-0.03 \pm 0.00}$ | | $0.02 \pm 0.00$ | | $0.00 \pm 0.00$ |
| **TWATER** | $\mathbf{-0.41 \pm 0.01}$ | | $0.03 \pm 0.00$ | | $0.06 \pm 0.00$ | |
| d$_t$ CAPE | | | | | | |
| Vh_SHEAR | | | | | | |
| T_2M | | | | | | |
| **AC_COS** | $-0.34 \pm 0.01$ | $\mathbf{-0.07 \pm 0.00}$ | $-0.06 \pm 0.01$ | $0.00 \pm 0.00$ | $0.09 \pm 0.00$ | $0.02 \pm 0.00$ |
| **AC_SIN** | $0.02 \pm 0.01$ | $0.01 \pm 0.00$ | $-0.09 \pm 0.00$ | $-0.01 \pm 0.00$ | $0.01 \pm 0.00$ | $0.01 \pm 0.00$ |