# Peer review of "Vertical profiles of wind gust statistics from a regional reanalysis using multivariate extreme value theory"

_Nonlinear Processes in Geophysics, 2019_

## Referee Comment (RC1) · Anonymous Referee #1 · 23 Dec 2019

This manuscript describes a methodology to predict vertical profiles of wind gusts based on a number of covariates that are taken from a reanalysis (which only includes wind gust diagnostics at 10 m). It is interesting, well written, and technically sound. My only major complaint is that the test setup used here is still several steps away from a setup that would be used in operational forecasting. While replacing the reanalysis data by forecast data is straightforward, it would be interesting to see how much skill over climatology is retained when forecast uncertainty is added to the uncertainty in the statistical model presented here, that links covariate information to wind gusts at various vertical levels. An even more interesting question is only briefly discussed in the conclusions: in how far can the model estimated here be transferred to other locations?

That question is highly relevant for practical application of this method since wind gust observations at several vertical levels like the ones used here are rare, and the model would have to be transferable for this methodology to provide wind gust predictions at a wider range of locations. That being said, I certainly understand the challenges involved in investigating this transferability, so I am not suggesting that adding this to the manuscript is mandatory.

Specific comments:

85-90: Since references are provided, I don't think it is necessary to restate the theorem here.

94: I would suggest to be slightly more precise and state explicitly that $G$ and $G_u$ are the CDFs of the respective distributions

99: Here and later, the terms non-homogenous and non-stationary are used in a somewhat sloppy way. Based on the conext I understand that the authors basically want to say that these parameters are non-constant, i.e. they depend on covariates. I find especially the term non-stationary confusing and misleading here (and in 127)

99: I also find it strange to refer to height as 'space'. To me, the term 'space' implies at least two dimensions.

105: What is 'generalized height', where is this defined? Or do you just want to say 'normalized height'?

136/137: I feel the term 'stationary' is again misused here, now in a different way. To me, 'stationary' is not synonymous with 'unconditional'. Did you mean to say 'climatological distribution'?

180: How exactly is this 5-h time window defined? 5-h before the time t? Or centered around t? Please clarify, because this also has implications on using this methodology in a forecast (i.e. forward in time) context.
218: How is lambda determined? Also via cross-validation? Presumably some sort of data driven routine must have been used because the results are typically quite sensitive to the strenght of regularization.

Language and typos:

272: appropriate what? Some word seems to be missing here

283: This sentence essentially repeats the statement of the previous sentence

292: May -> March

322: climatorlogical -> climatological

333: therefor -> therefore

---

## Referee Comment (RC2) · Anonymous Referee #2 · 11 Jan 2020

The authors present a novel approach to modelling hourly peak wind speed using a generalized extreme value (GEV) distribution with height and time dependent parameters. These GEV parameters are functions of several covariates from the COSMO-REA6 reanalysis. The presented results are convincing and the paper is well written. There are only minor inaccuracies requiring clarification:

P2,L44-46. Ensemble model output statistic is often referred as non-homogeneous regression, so I don't see any reason for treating the two notions here separately.

P4,L85. I don't see the reason of formulating Gnedenko's theorem. I would just cite it and define the GEV distribution.

P4,L105. What do the authors mean by "generalized height".

P10,L271. Please clarify the sentence "We conclude ... " as something seem to be missing here.

Typos:

P2,L47. "In order to generate" instead of "In order generate"

P3,L78. "We pre-select" instead of "We pre-selection"

P5,L138. "All scores are evaluated" instead of "All scores are evaluation"

P7,:181. "COSMO" instead of "COMSO"

P11,L33. "therefore" instead of "therefor"

P26,Figure 11. In the legend I would write "30% neutral (b), and 11% unstable (c) cases."

---

## Author Comment (AC1) · 16 Mar 2020

*We thank the reviewer for his/her helpful comments. Your suggestions are greatly appreciated and lead to an improvement of the article. In the following we respond point-by-point (in italic) to your comments (in normal font).*

[Figure]

**Anonymous Referee #1**

This manuscript describes a methodology to predict vertical profiles of wind gusts based on a number of covariates that are taken from a reanalysis (which only includes wind gust diagnostics at 10 m). It is interesting, well written, and technically sound. My only major complaint is that the test setup used here is still several steps away from a setup that would be used in operational forecasting. While replacing the reanalysis data by forecast data is straightforward, it would be interesting to see how much skill over climatology is retained when forecast uncertainty is added to the uncertainty in the statistical model presented here, that links covariate information to wind gusts at various vertical levels. An even more interesting question is only briefly discussed in the conclusions: in how far can the model estimated here be transferred to other locations? That question is highly relevant for practical application of this method since wind gust observations at several vertical levels like the ones used here are rare, and the model would have to be transferable for this methodology to provide wind gust predictions at a wider range of locations. That being said, I certainly understand the challenges involved in investigating this transferability, so I am not suggesting that adding this to the manuscript is mandatory.

*You indeed raise interesting questions. We discuss this question in the conclusions section as follows (334-343): "Our post-processing strategy is applicable to NWP forecasts without relevant changes, except for the selection of the covariates. Particularly, if applied to ensemble forecasts, additional predictors such as the predictive uncertainty, quantiles or probabilities for threshold exceedances as derived from the ensemble may be considered. For an example of how to include ensemble statistics into the post-processing see Wahl (2015). In Friederichs et al. (2018, F2018) a similar*

*approach is applied to COSMO-DE-EPS forecasts to predict 6 hourly maxima of 10m wind gusts. Although not really comparable (i.e. hourly maxima in this study – 6 hourly maxima in F2018; a variety of covariates as predictors in this study – wind variables only in F2018), they obtain a BSS for a 14.8 m/s threshold and a QSS for the 99% quantile of about 40%, respectively. The forecast lead time in their study is between 12 and 18 hours. This suggests that forecasts errors at lead times of about 1 day for 6 hourly maxima is small enough to obtain reasonable skill. The respective skill scores at the 10 m level in this study amount to about 24% for the BSS and about 53% for the QSS. The skill scores are comparable and suggest, that similar skill scores may be obtained at higher levels."*

*We absolutely agree that the question of transferability to other locations is very important. It is therefore at the top of our list of future tasks. We included some further thoughts on this in the conclusions section (351-355): "This may be tested using observations from other weather masts in the model region. However, difficulties may arise because even the observations on the different masts are processed differently or are made with different measuring instruments. Furthermore, different topography and other local parameters can introduce systematic biases. At other locations only measurements of the 10 m are available, and it would be of interest to assess how well are estimates of gust statistics at higher levels which are only based on observations at 10 m."*

*A thorough analysis of transferability is needed and would go far beyond the scope of this study.*

Specific comments:

85-90: Since references are provided, I don't think it is necessary to restate the theorem here.

*We agree with you and the second reviewer and removed the theorem. Instead we*

*now just give the definition of the GEV:*

*83-86: "The asymptotic cumulative distribution function (cdf) $G$ is defined by*
$$G(y; \mu, \sigma, \xi) = \exp\left(-\left[1 + \xi\left(\frac{y-\mu}{\sigma}\right)\right]^{-1/\xi}\right) \xi \neq 0$$
$$= \exp\left(-\exp\left[-\left(\frac{y-\mu}{\sigma}\right)\right]\right) \xi = 0,$$

*on $\{y : 1 + \xi(y - \mu)/\sigma > 0\}$, where $-\infty < \mu < \infty$, $\sigma > 0$ and $-\infty < \xi < \infty$. The parameters are denoted as location for $\mu$, scale for $\sigma$, and shape for $\xi$."*

94: I would suggest to be slightly more precise and state explicitly that G and $G_u$ are the CDFs of the respective distributions

*You're right. We reformulated the sentence to connect $G$ with $Y$ and $G_u$ with $Y_u$ more accurately:*

*88-90: " $G(y; \mu, \sigma, \xi)$ denotes the cdf of the uncensored variable $Y$, whereas the censored GEV (cGEV) $G_u$ represents the cdf of $Y_u$ and is given as $G_u(y; \mu, \sigma, \xi) = G(y; \mu, \sigma, \xi)$ if $y \geq u$ and $G_u(y; \mu, \sigma, \xi) = 0$ otherwise."*

99: Here and later, the terms non-homogeneous and non-stationary are used in a somewhat sloppy way. Based on the context I understand that the authors basically want to say that these parameters are non-constant, i.e. they depend on covariates. I find especially the term non-stationary confusing and misleading here (and in 127)

*We removed stationary and non-stationary, as it might be misleading. We now mostly refer to non-homogeneous and state whether its in height or time, since it is consistent with the terminology used in the introduction (non-homogeneous regression), which is in fact what we do.*

*94-96: "We thus assume that $Y(z, t)$ follows a cGEV with $G_u(y; \mu(z, t), \sigma(z, t), \xi(z, t))$, such that the parameters $\mu(z, t)$, $\sigma(z, t)$, $\xi(z, t)$ vary in both height and time. The temporal non-homogeneity (i.e. non-stationarity) is explained through $L$ covariates*

[Figure]

$C_l(t)$ *assuming a linear regression ansatz"*

*122-123: "The constant parameters $\mu_{0k}$ and $\sigma_{0k}$ are not penalized, and thus a large shrinkage parameter $\lambda$ results in a temporally homogeneous cGEV model."*

*132-133: "Our reference is the censored GEV with constant parameters estimated using the observed gusts at each mast level individually, refereed to as climatology."*

99: I also find it strange to refer to height as 'space'. To me, the term 'space' implies at least two dimensions.

*We agreed, and replace "space" by "height" or by "vertical":*

*94-96: See above.*

*99-100: "In order to be able to interpolate the parameters vertically, we approximate their height dependence "*

*112-113: "In order to assess the predictability in the vertical, an additional leave-one-out procedure is applied, where the layer to be predicted is withheld during the estimation procedure."*

105: What is 'generalized height', where is this defined? Or do you just want to say 'normalized height'?

*We agree that 'normalized height' is more appropriate and in addition define what we mean by it.*

*101-102: "where $\eta \in [0, 1]$ is a normalized height equal to 1 at 250 m and 0 at 10 m height."*

136/137: I feel the term 'stationary' is again misused here, now in a different way. To me, 'stationary' is not synonymous with 'unconditional'. Did you mean to say 'climatological distribution'?

*We rewrote the sentence to be more clear on the reference distribution:*

*132-133: See above.*

180: How exactly is this 5-h time window defined? 5-h before the time t? Or centered around t? Please clarify, because this also has implications on using this methodology in a forecast (i.e. forward in time) context.

*We used the two hours before and after the time $t$, so that the value is centered around $t$. The sentence now reads:*

*176-177: "We also include the variance of VMAX_10M over the period from 2 hours before to 2 hours after the respective analysis time ($Var_t$ VMAX_10M) as a covariate."*

218: How is lambda determined? Also via cross-validation? Presumably some sort of data driven routine must have been used because the results are typically quite sensitive to the strength of regularization.

*This is correct. We started with varying $\lambda$ and LASSO paths to find an accurate $\lambda$. To describe this, we added the following:*

*214-218: "The variable selection is performed using the LASSO including cross-validation, providing eleven sets of penalized regression coefficients. The value of $\lambda$ is determined by analysing the cross-validated LASSO path, which describes the changes of the regression parameters with respect to $\lambda$. The LASSO is very sensitive to $\lambda$. We chose $\lambda = 0.02 \times m$, where $m$ is the number of observations, since a larger $\lambda$ leads to an excessive penalization, while a smaller $\lambda$ accepts almost all covariates as relevant."*

Language and typos:

272: appropriate what? Some word seems to be missing here
*The word model is missing.*
*270-271: "We conclude that the Legendre model represents an appropriate model for*

*all layers."*

283: This sentence essentially repeats the statement of the previous sentence
*Right, so we skipped that sentence; check at line 282.*

292: May → March
*Yes, thanks!*

322: climatorlogical → climatological
*In order to be more clear on the term climatology, we now use here 'the observations 50%-quantile':*
*320-321:"The censoring threshold is defined as the observations 50%-quantile at each mast level, respectively."*

333: therefor → therefore
*Thanks!*

---

## Author Comment (AC2) · 16 Mar 2020

*We thank the reviewer for his/her helpful comments. Your suggestions are greatly appreciated and lead to an improvement of the article. In the following we respond point-by-point (in italic) to your comments (in normal font).*

**Anonymous Referee #2**

The authors present a novel approach to modelling hourly peak wind speed using a generalized extreme value (GEV) distribution with height and time dependent parameters. These GEV parameters are functions of several covariates from the COSMO-REA6 reanalysis. The presented results are convincing and the paper is well written. There are only minor inaccuracies requiring clarification:
*Thanks!*

P2,L44-46. Ensemble model output statistic is often referred as non-homogeneous regression, so I don't see any reason for treating the two notions here separately.
*You are right. We changed the sentence accordingly. It now reads:*
*44-46: "Probabilistic methods employ non-homogeneous regression, e.g., Thorarinsdottir and Johnson (2012) for wind gusts, and Lerch and Thorarinsdottir (2013), Scheuerer and Möller (2015), or Baran and Lerch (2015) for wind speed."*

P4,L85. I don't see the reason of formulating Gnedenko's theorem. I would just cite it and define the GEV distribution.
*We agree with you and the first reviewer and removed the theorem. Instead we now just give the definition of the GEV:*
*83-86: "The asymptotic cumulative distribution function (cdf) $G$ is defined by*
$$G(y; \mu, \sigma, \xi) = \exp\left(-\left[1 + \xi\left(\frac{y-\mu}{\sigma}\right)\right]^{-1/\xi}\right) \xi \neq 0$$
$$= \exp\left(-\exp\left[-\left(\frac{y-\mu}{\sigma}\right)\right]\right) \xi = 0,$$

*on $\{y : 1 + \xi(y - \mu)/\sigma > 0\}$, where $-\infty < \mu < \infty$, $\sigma > 0$ and $-\infty < \xi < \infty$. The parameters are denoted as location for $\mu$, scale for $\sigma$, and shape for $\xi$."*

P4,L105. What do the authors mean by "generalized height".
*We agreed that this was misleading. We now use the term 'normalized height' and added its definition:*
*101-102: "where $\eta \in [0, 1]$ is a normalized height equal to 1 at 250 m and 0 at 10 m height."*

P10,L271. Please clarify the sentence "We conclude ... " as something seem to be missing here.
*The word model is missing.*
*270-271: "We conclude that the Legendre model represents an appropriate model for all layers."*

Typos:

P2,L47. "In order to generate" instead of "In order generate"
*Thanks!*

P3,L78. "We pre-select" instead of "We pre-selection"
*Thanks!*

P5,L138. "All scores are evaluated" instead of "All scores are evaluation"
*Thanks!*

[Figure]

P7,:181. "COSMO" instead of "COMSO"
*Thanks, so we changed here (and lines before, where we made the same mistake).*

P11,L33. "therefore" instead of "therefor"
*Thanks!*

P26,Figure 11. In the legend I would write "30% neutral (b), and 11% unstable (c) cases.
*Yes, '(c)' is missing here, so we changed:*
*Figure 11: "According the LI, the data are classified in 53 % stable (a), 36 % neutral (b), and 11 % unstable cases (c)."*